

Laboratory and field evaluation of the Aerosol Dynamics Inc. concentrator (ADIc)
for aerosol mass spectrometry
Sanna Saarikoski[1], Leah R. Williams[2], Steven R. Spielman[3], Gregory S. Lewis[3], Arantzazu
Eiguren-Fernandez[3], Minna Aurela[1], Susanne V. Hering[3], Kimmo Teinilä[1], Philip Croteau[2], John
T. Jayne[2], Thorsten Hohaus[2,+], Douglas R. Worsnop[2], Hilkka Timonen[1]
[1] Atmospheric Composition Research, Finnish Meteorological Institute, Helsinki, Finland
[2] Center for Aerosol and Cloud Chemistry, Aerodyne Research, Inc., Billerica, MA, USA
[3] Aerosol Dynamics Inc., Berkeley, CA, USA
[+] Now at Institute of Energy and Climate Research, IEK-8: Troposphere, Forschungszentrum
Juelich GmbH, Juelich, Germany



**Abstract**
An air-to-air ultrafine particle concentrator (Aerosol Dynamics Inc. concentrator; ADIc) has been
designed to enhance on-line chemical characterization of ambient aerosols by aerosol mass
spectrometry. The ADIc employs a three-stage, moderated water-based condensation growth tube
coupled to an aerodynamic focusing nozzle to concentrate ultrafine particles into a portion of the
flow. The system can be configured to sample between 1.0–1.7 L min$^{-1}$ with an output concentrated
flow between 0.08–0.12 L min$^{-1}$, resulting in a theoretical concentration factor (sample flow/output
flow) ranging from 8 to 21. Laboratory tests with monodisperse particles show that the ADIc is
effective for particles as small as 10 nm. Laboratory experiments conducted with the Aerosol Mass
Spectrometer (AMS) showed no shift in the particle size after the ADIc, as measured by the AMS
particle time-of-flight. The ADIc-AMS system was operated unattended over a one-month period
near Boston, Massachusetts. Comparison to a parallel AMS without the concentrator showed
concentration factors of $9.7 \pm 0.15$ and $9.1 \pm 0.1$ for sulfate and nitrate, respectively, when operated
with a theoretical concentration factor of $10.5 \pm 0.3$. Concentration factor of organics was lower,
possibly due to the presence of large particles from nearby road-paving operations, and a difference
in aerodynamic lens cutoff between the two AMS instruments. Another field deployment was
carried out in Helsinki, Finland. Two ~10-day measurement periods showed good correlation for
the concentrations of organics, sulfate, nitrate and ammonium measured with an Aerosol Chemical
Speciation Monitor (ACSM) after the ADIc, and a parallel AMS without the concentrator.
Additional experiments with an AMS alternating between the ADIc and a bypass line
demonstrated that the concentrator did not change the size distribution or the chemistry of the
ambient aerosol particles.



## 1 Introduction

Particles in the ambient atmosphere are of concern for human health, air quality and climate change (Pope and Dockery, 2006; Lelieveld et al., 2015; IPCC 2014). Measurement of the chemical characteristics of particles, and the health effects associated with their inhalation, often benefit from higher sample load which can be achieved by increasing sample flow rate, extending sampling time or using a particle concentrator. Enrichment of particle number or mass concentration is particularly important for measurements in regions where particle concentrations are low, such as in Arctic or Antarctic background areas (10–1000 particles per $cm^{-3}$, Asmi et al., 2010; Tunved et al., 2006). An increase in particle mass can also benefit the measurement of trace aerosol components such as metals, or improve the determination of chemically resolved size distributions.

Several air-to-air concentrators have been designed to increase the concentration of particles with respect to the suspending gas volume, and to thereby providing enhanced aerosol detection. To be beneficial, the concentrator should be small, easy to maintain and capable of operating several days or even weeks unattended. Even more importantly, the concentrator should provide stable enrichment of particles, and maintain aerosol chemical and physical and properties such as composition and size distribution. Virtual impactors are a well-known type of air-to-air particle concentrators that use a low-velocity sampling probe to sample a particle flow exiting from a nozzle but they are typically ineffective for the submicrometer (< 1 μm) and ultrafine (< 100 nm) particle size ranges that are of most interest for atmospheric and health-related particle studies. Current air-to-air concentrators for small particles couple condensational growth with traditional virtual impactors, e.g., the Versatile Aerosol Concentration Enrichment System (VACES, Kim et al., 2001), the miniature VACES (Geller et al., 2006; Saarikoski et al., 2014) or the Harvard Ultrafine Concentrated Ambient Particle System (HUCAPS, Gupta et al., 2004). However, these systems are ineffective for particles below ~30 nm in diameter. Moreover, with long condensational growth times, these approaches have been shown to feature the undesirable effect of changing the particle chemical composition (e.g., Saarikoski et al., 2014).

Here we present a new air-to-air particle concentrator, the Aerosol Dynamics Inc. concentrator (ADIc), that is based on the three-stage, laminar-flow, water-based condensational growth approach used in the Sequential Spot Sampler (Eiguren Fernandez et al., 2014; Pan et al., 2016),





and in some water condensation particle counters (CPCs, Hering et al., 2017; 2018). This system
is designed specifically for instruments with low sampling flow rates on the order of 0.1 L min$^{-1}$.
It offers concentration factors (CFs) of 8 to 21 for particles as small as 10 nm diameter in an output
flow that is noncondensing at typical room temperatures (i.e. with dew points below 16 °C).
Previously, a preliminary version of this concentration approach that used a two-stage growth tube
was coupled to an Aerosol Time-of-Flight Mass Spectrometer (ATOFMS, Zauscher et al., 2011)
and showed both concentration enhancement and lack of chemical artifacts. However, this
preliminary system was not stable enough for long-term operation.
The three-stage growth column version of the ADIc described here eliminates excess water vapor
in the output flow and decreases the residence time for the particle in the droplet phase, with the
objective of minimizing chemical artifacts as well as providing long-term stability. The ADIc is a
smaller scaled version of the approach used in the nano-particle charger reported by Kreisberg et
al. (2018), for which chemical artifacts, evaluated using Thermal Desorption Chemical Ionization
Mass Spectrometry, were found to be mostly insignificant. The ADIc is tailored for use with an
aerosol mass spectrometer, such as the Aerodyne Aerosol Mass Spectrometer (AMS) or ATOFMS.
In this paper, the ADIc was evaluated in laboratory experiments that explored its influence on
particle size and chemical composition. The ADIc was also evaluated in field measurements
conducted in two different environments (urban and urban background) and with different
commonly used types of aerosol mass spectrometers. Moreover, long term (weeks to months)
unattended operation of the ADIc was demonstrated.

**2 Experimental**
**2.1 System description of the ADIc**
The ADIc uses a laminar flow, water- based condensation growth tube coupled to an aerodynamic
focusing nozzle to provide concentration of particles from a 1–1.7 L min$^{-1}$ sample flow into a 0.08–
0.12 L min$^{-1}$ concentrated output flow. This system uses a three-stage moderated aerosol
condensation approach (Hering et al., 2014) whereby the aerosol flow passes through a wet-walled
tube with three distinct temperature regions (Fig. 1). In the first stage, the conditioner has cold
walls and brings the flow to known conditions of cool temperature and high relative humidity
(RH). The second, initiator stage, has warm walls and provides the water vapor that creates the



supersaturation for particle activation, while the last, cool-walled moderator stage provides time
for particle growth while simultaneously removing water vapor from the flow. The water vapor
saturation level reaches a value of 1.4 in the initiator while maintaining temperatures below 30 °C
in the majority of the sample flow, and simultaneously providing for output flow dew points below
16 °C. Thus, the water vapor content of the output flow is reduced to typical ambient conditions,
making it easier to handle, and minimizing the amount of water reaching the detection system. The
wetted walls are maintained by a single wick formed from rolled membrane filter media and the
flow is laminar throughout the ADIc system.
Within the growth tube, particles with diameters above 5–10 nm are activated and grow by
condensation to form droplets of approximately 1.5–4 µm in diameter. The cooled, droplet-laden
flow passes through a 1-mm diameter nozzle wherein the droplets are aerodynamically focused
along the central core of the flow, much as described by Fuerstenau et al. (1994). The ADIc
contains an annular slit in the side wall of this nozzle, through which the majority (85–95 %) of
the flow (discard flow) is extracted. The remaining 5–15 % of the flow contains the droplets which
have been focused aerodynamically. Water evaporates from the droplets once the flow regains
ambient (20–25 °C) temperature to provide a concentrated aerosol flow (output flow). The system
is designed to minimize the time the particle is a droplet, with the objective of minimizing chemical
artifacts, similar to the nano-particle charging system (Kreisberg et al., 2018).
The exact design of the focusing and flow extraction nozzle is based on numerical modeling done
using the Comsol Multiphysics package. Numerical modeling results, presented in Fig. S1 for the
final design, show that particles smaller than 1µm follow the gas flow trajectories and are extracted
through the annular slit while those above 6 µm over-focus and collide with the opposite wall.
However, intermediately sized particles, corresponding to a Stokes number (St) of 0.5 to 3.5, are
aerodynamically focused in the region near the centerline of the flow. These particles follow the
remaining flow, the output flow, which continues straight, thus providing a concentrated flow for
sampling with aerosol instrumentation. The theoretical concentration factor is determined by the
ratio of the sample flow rate to the output flow rate and can be varied between 8 and 21.
Two prototype concentrators (Prototype 1 and 2) were used in this study, both having the same
dimensions for the growth tube and nozzle. The conditioner, initiator and moderator are 140 mm,
51 mm and 102 mm long, respectively, separated by 7.5 mm thick insulator sections. In both





prototypes the growth tube was lined with a 9 mm-ID, ~1.5 mm-thick wick formed from rolled
membrane filter. The conditioner and moderator were cooled using Peltier heat pumps and the
initiator and focusing nozzle were heated resistively. All three regions used proportional-integral-
derivative (PID) control to maintain set-point temperatures. Distilled water was injected into the
initiator stage at a rate of 5 µL min$^{-1}$ and excess water was removed from the base of the wick
carried by a small flow of ~0.05 L min$^{-1}$ of air into a waste bottle. Other than packaging, the only
difference between the prototypes was that Prototype 1 had a mass flow meter to measure the
discard flow while Prototype 2 did not have this option. The theoretical CF for Prototype 1 was
determined continuously from the measured flows, while for Prototype 2 the theoretical CF was
determined from the sample and concentrated flow rates measured before and after each
experiment. The size of the ADIc is approximately 30 x 30 x 50 cm (W x D x H) and the weight
is ~11 kg.

**2.2 Evaluation in the laboratory**
**2.2.1 Particle number measurements at ADI**
The performance of the ADIc for particle counting was evaluated in the laboratory at Aerosol
Dynamics Inc. (ADI) using monodisperse particles generated by atomization, followed by drying
and charge conditioning (soft X-ray, Model 3087, TSI Inc., Shoreview, US). Particles were size
selected using a nano-differential mobility analyzer (DMA, Model 3085, TSI Inc., Shoreview, US)
for sizes between 5 nm and 60 nm and using the Aerosol Dynamics Inc. high-flow DMA
(Stolzenburg et al., 1998) for sizes between 20 nm and 600 nm. Particle concentrations were
measured in the sample flow and in the concentrated output flow using water-based CPCs.
Prototype 1 was evaluated with mono-mobility ammonium sulfate (AS) particles with a pair of
prototype Model 3785 (TSI Inc., Shoreview, US) water-based CPCs and a Model 3783 CPC (TSI
Inc., Shoreview, US) to simultaneously measure particle concentrations in the sample flow, in the
discard flow, and in the concentrated output flow, respectively. The sample flow was fixed at 1.0
L min$^{-1}$, and the output flow was 0.12 L min$^{-1}$ (theoretical CF = 8.3). The operating temperatures
for conditioner (Tcon), initiator (Tini), moderator (Tmod) and focusing nozzle (Tnoz) were 5, 26,
10 and 30 °C, respectively (see Table 1).





Similar evaluation experiments were carried out on Prototype 2 but its operation was tested under
two flow regimes. First, experiments were done at 1.0 L min⁻¹ sample flow and 0.11 L min⁻¹ output
flow (theoretical CF = 9.1), with similar operating temperatures to Prototype 1. To test higher CFs,
experiments were also done at a sample flow rate of 1.5 L min⁻¹ and an output flow of 0.11
L min⁻¹ for a theoretical CF of 13.6. The growth tube is sized for low-flow operation, such that the
centerline supersaturation reaches its maximum at the end of the warm initiator section. At the
higher flow rate, the residence time is shorter, and thus for the same operating temperatures the
peak supersaturation is lower. To compensate, the initiator was operated at a warmer wall
temperature, thereby providing a similar value for the calculated peak super-saturation. The
operating temperatures for the high flow were Tcon = 6 °C, Tini = 31 °C, Tmod = 8 °C, and Tnoz
= 35 °C (Table 1).
In addition to laboratory generated AS particles, both prototypes were tested with laboratory air
using a pair of water-based CPCs, one sampling upstream of the ADIc and one sampling
downstream.

### 2.2.2 Particle chemistry at ARI and FMI

The performance of the ADIc in terms of particle chemistry was evaluated at Aerodyne Research,
Inc. (ARI) and at the Finnish Meteorological Institute (FMI). Laboratory experiments were carried
out by using particles generated with a constant output atomizer (Model 3076, TSI Inc., Shoreview,
US) from AS or ammonium nitrate (AN) in deionized water, or from dioctyl sebacate (DOS) in 2-
propanol. Generated particles were dried with a silica gel dryer and the desired monodisperse
particle size fraction was selected using a DMA (Model 3080, TSI Inc., Shoreview, US). A valve
system was used to alternate between passing the particles through the ADIc and bypassing it.
Temperature and flow settings used in the ADIc during the ARI and FMI experiments are given
in Table 1.
Particle size and chemical composition were measured with several different versions of the AMS,
including a high-resolution time-of-flight aerosol mass spectrometer (HR-AMS, Aerodyne
Research Inc., Billerica, US; DeCarlo et al., 2006), a soot-particle aerosol mass spectrometer (SP-
AMS, Aerodyne Research Inc., Billerica, US; Onasch et al., 2012), a quadrupole aerosol mass



spectrometer (Q-AMS, Aerodyne Research Inc., Billerica, US; Canagaratna et al., 2007) and a
quadrupole aerosol chemical speciation monitor (ACSM, Aerodyne Research Inc., Billerica, US;
Ng et al., 2011). These instruments all operate on the same principle. Aerosol particles are sampled
through an aerodynamic lens, forming a narrow particle beam that is transmitted into the detection
chamber where the non-refractory species are flash vaporized upon impact on a hot surface (600
°C). The particle vapor is ionized using electron impact ionization (70 eV) and detected by the
mass spectrometer. Particle size (particle time of flight (PToF) data) is determined from particle
flight time in the vacuum chamber after passing through a chopper. The typical size range of
particles detected with an AMS is 70 nm to 700 nm (Liu et al., 2007). In addition to the thermal
vaporizer, the SP-AMS incorporates an intracavity Nd-YAG (1064 nm) laser that enables the
determination of refractory black carbon (rBC) and metal containing particles (Onasch et al., 2012;
Carbone et al., 2015). The ACSM does not include particle size measurement capability.
HR- and SP-AMS data was analyzed with the Squirrel (v1.57H)/Pika (v1.16H) and Squirrel
(v1.60P)/Pika (v1.20P) analysis package, respectively. Additionally, high resolution (HR) size
distribution data from the SP-AMS was analyzed with Squirrel (v1.62A)/Pika (v1.22A) package.
Both the HR-AMS and SP-AMS instruments were equipped with a multiplex chopper and the
measured size distributions were normalized to the mass spectra. Q-AMS data was analyzed with
AMS Analysis Toolkit 1.43. ACSM data was analyzed with ACSM Local (v1.6.1.1). All of the
analysis software runs in the Igor 6 (WaveMetrics, Inc.) programming environment. The three
AMS instruments and the ACSM were calibrated for ionization efficiency (IE) of nitrate and
relative ionization efficiency (RIE) of both ammonium and sulfate, using size selected single
component particles of AN or AS (Budisulistiorini et al., 2014).

### 208    2.3 Field testing

The ADIc was tested for ambient aerosol at two different locations. At ARI, particles were sampled
from a roof top sampling station on the ARI building at 45 Manning St., Billerica, MA (42.53, -
71.27, 60 m a.s.l.), located about 60 m NE of a major freeway. Ambient air was sampled at 3 L
min$^{-1}$ through a 2.5 μm cut cyclone and split between two paths. The first path went to an HR-
AMS and a CPC (Model 3776, TSI Inc., Shoreview, US). The second path went to the ADIc
followed by a Q-AMS and a CPC (Model mCPC, Brechtel, Hayward, US). Two valves allowed





the ambient air to bypass the ADIc and directly enter the Q-AMS. Both AMSs recorded data at 2-
minute time resolution. Ambient sampling was conducted from 1 to 26 August 2014. The default
collection efficiency (CE) of 0.5 for ambient particles was applied to data from both AMS
instruments. Local ambient temperature was downloaded from Weather Underground for station
KMABILLE10 and ambient RH data was downloaded from NOAA for Hanscom.
The second ambient sampling location was at an urban background station (SMEARIII; Station
for Measuring Ecosystem-Atmosphere Relationships, 60.20, 24.95, 30 m a.s.l., described by Järvi
et al., 2009) located at the Kumpula campus near the FMI building, about 5 km NE of the Helsinki
city center, Finland. The station is surrounded by office buildings on one side and a small forest
and botanical garden on the other side. Ambient particles were sampled through a 2.5 μm cyclone
with a flow rate of 3 L min$^{-1}$. Sample flow was split into two sampling lines; the first line went to
the SP-AMS (with an additional bypass flow of 1.3–2 L min$^{-1}$) and the second line to the ADIc
followed by an ACSM. The ACSM data was averaged approximately to 10-minute time resolution
(10 times open + close, m/z range: 10–150, scan rate 200 ms/amu) and the SP-AMS measured
with a time resolution of 1.5 minutes. Two sample flow regimes were tested with the ACSM+ADIc
system; the sample flow was set to either 1.7 L min$^{-1}$ or 1.0 L min$^{-1}$ while the output flow of the
ADIc was determined by the ACSM inlet flow of 0.08 L min$^{-1}$, giving a theoretical CF of 21.3 and
12.5 for high and low sample flow, respectively. Additionally, in a separate set of experiments, the
ADIc was installed upstream of the SP-AMS in order to investigate the influence of the ADIc on
high resolution mass spectra and size distributions. Those tests were carried out in the high flow
regime (theoretical CF of 21.3) in order to maximize the increase in HR organic and rBC mass
spectral and PToF signals with the ADIc. The SP-AMS measurements were conducted by
switching the laser on and off. Laser off data was utilized when the SP-AMS was compared with
the ACSM+ADIc and laser on data was used for the period when the ADIc was installed in front
of the SP-AMS. The default CE of 0.5 for ambient particles was applied to both ACSM and SP-
AMS data. An RH sensor was installed in the ACSM line after the ADIc. Ambient meteorological
parameters were recorded at the Kumpula Weather station. Field measurements at SMEAR III
were conducted between 13 July to 22 October 2018, with sampling on about 27 different days.
Temperature settings of the ADIc during the field campaigns at ARI and FMI are given in Table

244    1.



## 3 Results and discussion

### 3.1 Laboratory evaluation

#### 3.1.1 Concentration factor

Figure 2 shows laboratory results for monodisperse AS particles for two flow regimes. The
measured concentration factor, defined as the ratio of particle number concentration in the output
flow of the ADIc to that in the sample flow, is plotted as a function of particle mobility diameter.
Data for the lower flow regime is from Prototype 1, which was subsequently tested at ARI for
aerosol chemical species. For the lower flow, the average measured CF was $7.7 \pm 0.3$ for the
particles larger than 15 nm, compared to a theoretical CF of 8.3. Data shown for the higher flow
regime was obtained with Prototype 2, which was later tested at FMI for particle chemistry and
size distributions. For the higher flow, the measured CF was $11.9 \pm 0.2$, compared to a theoretical
CF of 13.6, for 50–305 nm particles. When operated in the lower flow regime, Protoype 2 data is
similar to that for Prototype 1, with a measured CF of $7.0 \pm 0.5$ (data not shown). The influence of
ADIc on particle size was investigated in more detail with aerosol mass spectrometers (Sect.

259    3.1.2.).

The ratio of measured to theoretical CF was ~0.9 (see Table 2), suggesting that 90 % of the
particles in the sample flow were focused into the output concentrated flow. In the experiments
conducted on Prototype 1, the particle concentration was also measured in the discard flow, and it
accounted for $9 \pm 2$ % of the sampled particle concentration at sizes above 20 nm, on average. The
fraction of particles in the discard flow showed a small, but systematic, dependence on particle
size with the fraction decreasing from 12 % at 18 nm to 6 % at 600 nm. The unaccounted particles
(2 % on average) were presumably lost in the transport lines or in the focusing nozzle itself.
To evaluate the stability of the ADIc, both prototypes were operated for several days while
sampling laboratory air. Particle number concentrations were measured in the sample flow and in
the output flow. Particle concentration varied between 900 and 15000 # $cm^{-3}$. For the lower flow
regime data (Fig. S2a–b), the measured CF was of $5.7 \pm 0.4$ with the theoretical CF of 7.5. Linear
regression of that data yielded a correlation coefficient ($R^2$) of 0.984. In the higher flow regime
(Fig. S2c–d), the measured CF was $9.0 \pm 0.7$, with a theoretical CF of 13.6. For that data the
correlation coefficient ($R^2$) was 0.940. It is important to note that particle concentrations were





measured using CPCs with a 5 nm activation threshold while the ADIc threshold is closer to 10
nm. Thus, particles below 10 nm in the ambient size distribution would not be concentrated,
leading to a lower measured CF and a lower ratio of measured/theoretical CF than in Table 2.  In
addition, changes in the ambient size distribution can lead to some variability in the measured CF.
Importantly, no systematic change was observed throughout the experiments.

**3.1.2 Chemical composition and particle size**
The dependence of CF on particle chemical composition was evaluated in the laboratory with size-
selected 300 nm AS and AN particles and a subsequent analysis of concentrated aerosol by an HR-
AMS. The theoretical and the measured CF for ammonium and sulfate from AS and for ammonium
and nitrate from AN are given in Table 2. Compared to CF obtained for particle number
concentration, the ratio of measured to theoretical CF was the same for AS while for AN the
measured CF was slightly closer to the theoretical CF.
The influence of the ADIc on particle size was investigated by using monodisperse AS, AN and
DOS particles in the size range of 30 to 340 nm (mobility diameter). Size and chemical
composition of particles with and without the ADIc were analyzed by an SP-AMS. Measurements
were carried out in the high flow regime (theoretical CF of 21.3). Figure 3 shows the vacuum
aerodynamic diameter ($d_{va}$) for sulfate (from AS), nitrate (from AN) and organics (from DOS) as
measured for concentrated versus unconcentrated aerosol. The regression slope was 1.02, the
intercept was -2.51, and the correlation coefficient ($R^2$) was 0.999 showing that the particle
diameter was not changed by passing through the ADIc for any of the measured particle sizes or
chemical species.

**3.2 Field Evaluation**
**3.2.1 Ambient organics and rBC**
The performance of the ADIc for ambient aerosol was examined at two locations; at a roof top
sampling station on the ARI building and at SMEAR III in Helsinki. In order to investigate the
impact of the ADIc on aerosol organic and rBC chemistry, the SP-AMS was installed behind the



ADIc at SMEAR III and measured alternately from the output flow of the ADIc and a bypass line.
Measurements were performed on 11 different days in June, July and August 2018 with a total
sampling time of ~7 hours behind the ADIc and ~7 hours in bypass. Average high-resolution mass
spectra for organics and rBC with and without the ADIc are presented in Fig. 4. In general, organics
at SMEAR III were highly oxygenated with large oxygen to carbon ratio (O:C) and large organic
carbon to organic matter ratio (OC:OM). The elemental composition of organics did not change
noticeably when the sample was passed through the ADIc. The correlation between the mass
spectral ions with and without the ADIc for each fragment family are presented in Fig. 4 c–f. The
correlation was uniformly high ($R^2 > 0.987$) and the slope describing the measured CF was on
average $19.2 \pm 3.2$. The slope was smallest for the most oxygenated fragment family $C_xH_yO_{z,\, z>1}$
and largest for $C_x$ (rBC) and was smaller than theoretical CF (21.3) for all families except the $C_x$
family. Smaller measured than theoretical CF is in agreement with the results obtained in the
laboratory tests (see Table 2) while the reason for a larger measured than theoretical CF for $C_x$ is
still unclear. Overall, based on these tests, it can be concluded that passing through the ADIc does
not significantly change the fragmentation or the elemental composition of organics in the ambient
particles.

**3.2.2 Mass size distributions**
The SP-AMS data with and without the ADIc was also used to investigate the impact of the ADIc
on particle mass size distributions. Figure 5 compares the mass size distribution for organics,
sulfate, nitrate and ammonium sampling through the ADIc and sampling from the bypass line. The
PToF data was collected and analyzed in unit mass resolution (UMR) mode. Figure 5 demonstrates
that the size distribution of ambient aerosol particles was not affected by passing through the ADIc.
In addition, Fig. 5d shows significant improvement in signal to noise for ammonium when
concentrating the sample flow.
Additional SP-AMS size distribution data was collected and analyzed in HR mode on one day with
a total sampling time of 70 minutes in bypass and 70 minutes through the ADIc. HR size
distributions are shown in Fig. 6 for major chemical species and for several specific fragment ions.
The much higher signal to noise in the concentrated PToF traces gives better chemical resolution
of the size distribution. The bimodal size distribution for organics is clear in the ADIc data in Fig.



6a with hydrocarbon-like fragments (e.g., $C_3H_7$ and $C_4H_9$ in Fig. 6h and 6k) contributing to the
mode at $d_{va}$ = 160 nm and more oxygenated fragments (e.g., $C_2H_3O$, $CO_2$, $C_2H_4O_2$ and $C_3H_5O$ in
Fig. 6g, 6i, 6j and 6l) contributing to the mode at $d_{va}$ = 400 nm. In addition, the higher signal to
noise in the concentrated sample enables PToF measurement for very small signals such as
chloride (Fig. 6e) or $CO_2$ (Fig. 6i) and improves the PToF measurement for smaller signals such
as rBC (Fig. 6f).

### 3.2.3 Long-term Stability

The long-term operation of the ADIc was tested at ARI where it ran for more than three weeks
without user maintenance or intervention. The measured CFs from comparing the Q-AMS mass
loading to the HR-AMS mass loading are presented in Fig. 7 with the average values presented in
Table 3. The theoretical CF was calculated from the ADIc discard flow rate and the Q-AMS inlet
flow rate (equal to ADIc outlet flow) as theoretical CF = (discard flow + Q-AMS inlet flow)/Q-
AMS inlet flow. Discard and Q-AMS flows were logged in real-time. The slight variation in
theoretical CF was due to variations in the Q-AMS inlet flow rate, not variations in the discard
flow. The gap in the data between 21 and 23 August 2014 was due to an issue with the HR-AMS,
not with the ADIc.
The measured CFs for nitrate and sulfate were 85 to 90 % of theoretical CFs, consistent with the
laboratory measurements presented in Table 2. The measured CF for ammonium was higher than
the theoretical value which may indicate that the aqueous droplets in the ADIc initiator and
moderator stages absorbed gas-phase ammonia that remained in the particles after drying. This
effect has been observed for acidic particles in the miniature VACES (Saarikoski et al., 2014). The
ambient aerosol in this study was possibly slightly acidic with an average ratio of measured to
predicted ammonia of 0.9 ± 0.15 in the HR-AMS data. Another possibility is that the RIE for
ammonium was incorrect for one or both of the instruments, even though it was measured three
times during the experiment. This is supported by the fact that the measured CF was greater than
one during periods when the Q-AMS was bypassing the ADIc (Table 3).
The measured concentration factor (6.1 ± 0.8) for organics was much lower than the theoretical
value (10.5 ± 0.3).  This was caused by a difference in the cutoff of the aerodynamic lenses in the



two AMS instruments. During this time period, organics were dominated by emissions from road
paving activities which generate large, hydrocarbon-like particles. Figure S3 shows the size
distributions for organics, mass-to-charge ratio (m/z) 44 and m/z 57 for the HR-AMS and the Q-
AMS+ADIc. It is clear that the size distributions for organics and m/z 57 from the Q-AMS were
missing mass above $d_{va}$ ~ 700 nm that was measured by the HR-AMS, leading to a lower measured
CF for organics. The m/z 44 size distributions, representative of accumulation mode aerosol
particles, were similar in the two instruments because the mass of m/z 44 was below the lens cutoff.
The measured CF for m/z 44 in Fig. S3b was 9.2 while the measured CF for m/z 57 in Fig. S3c
was only 3.9. The measured CF for organics also showed a larger diurnal variation than the
measured CFs for the other species (Fig. 7), likely because road paving activities took place at
night leading to a lower measured CF at night-time.

**3.2.4 Concentrating under high and low flow regimes**
The performance of the ADIc with ambient aerosol was also tested systematically under two flow
regimes. Although the growth tube in the ADIc is sized for low-flow operation, in some cases it
can be beneficial to operate the ADIc with the largest possible CF, for example, when very small
signals (e.g,. metals, PToF) are of interest, or the ambient concentrations are extremely low. High
(1.7 L min$^{-1}$) and low (1.0 L min$^{-1}$) sample flows, resulting in theoretical CFs of 21.3 and 12.5,
respectively, were investigated at SMEAR III with the ADIc installed in front of an ACSM while
the SP-AMS was sampling from the bypass line. The data from the ACSM+ADIc was corrected
for the CF by dividing the concentrations by 0.9 * theoretical CF since the laboratory tests and the
field campaign at ARI suggest that the measured CF is likely to be 90 % of the theoretical CF.
The time series of all chemical species measured with the ACSM+ADIc and SP-AMS track each
other well and the average mass loadings agreed within 20–30 % (Fig. 8), within the estimated
uncertainty of 34–38 % for AMS measurements (Bahreini et al., 2009). In the high flow regime,
the corrected ACSM+ADIc mass loadings were systematically higher for organics, sulfate and
ammonium compared to the SP-AMS. This might be caused by the lack of simultaneous
measurement of the sample flow rate, so that any error in the sample flow rate before/after the
experiment could propagate into the theoretical CF and thus into the correction factor. For nitrate,
the corrected ACMS+ADIc mass loading varied above the SP-AMS during the afternoon and





below during the night. Under low flow conditions, there was a time period of about 12 hours on 18 and 19 September when the corrected ACSM+ADIc mass loadings for nitrate and chloride were much lower than corresponding mass loadings from the SP-AMS. During this period, the aerosol particles were also not neutralized (i.e., measured ammonium was lower than ammonium predicted from the measured anions). Based on the ratio of m/z 46 to m/z 30, nitrate was in the form of inorganic nitrate (e.g., $NH_4NO_3$) rather than organic nitrates. The reason for the lower concentrations of nitrate and chloride with the ACSM+ADIc during this 12 hour period is not clear.

The relative humidity was measured after the ADIc near the Q-ACSM inlet. RH was relatively constant at $63 \pm 6$ %, consistent with a dewpoint of 16 °C at the outlet of the ADIc and a room temperature of about 25 °C. This was somewhat higher than the recommended operating RH of 20–40 % for AMS/ACSM instruments, but not high enough to cause an increase in the collection efficiency (Middlebrook et al., 2012). However, using a dryer in between the ADIc and the AMS/ACSM would reduce any potential uncertainty due to RH affecting CE.

In terms of Q-ACSM measurement, a particularly important improvement in signal to noise with the ADIc was achieved. Figs. 9a and 9b show 30-minute time resolution data collected with the Q-ACSM without the ADIc, and Figs. 9b and 9d display 10-minute time resolution data collected with the Q-ACSM+ADIc for ammonium and *m/z* 60, a tracer m/z for biomass burning. Compared to the SP-AMS data averaged to the same time resolution, it is evident that the signal to noise for the concentrated Q-ACSM data is similar to the SP-AMS. As a consequence, use of the ADIc with the ACSM will improve determination of ammonium and thus provide better estimates of particle neutralization and CE for ambient aerosol. In addition, better signal to noise for tracer m/z's will improve source apportionment with statistical methods such as positive matrix factorization (PMF).

**4 Conclusions**

The ADIc is tailored for the low (~0.08 L min$^{-1}$) inlet flow of aerosol mass spectrometers such as the AMS and ACSM and provides a factor of 8–21 enrichment in the concentration of particles. This concentration factor depends primarily on the ratio between the sample flow and the output



flow, and is found to be independent of particle size above about 10 nm. The system is relatively
small, and easily interfaced with the AMS.
Particle chemical composition and particle size measured with an SP-AMS were not affected by
the condensational growth and evaporation process in the ADIc. Moreover, the ADIc ran
unattended for a period of almost one month at a field site. Measured concentration factors for
ambient aerosol particles in two different locations showed some variation that is not fully
understood. However, the ADIc provides improved detection of low signals that outweighs a slight
increase in uncertainty in the mass loadings. Improved detection limits will be important especially
in remote areas where particle concentrations are low, and for measuring size distributions that
typically need longer averaging periods. Additionally, use of the ADIc will be important for
improving source apportionment with Q-ACSM data by gaining better time-resolution and/or
signal to noise ratio.

*Data availability.* Data presented in this article is available upon request.

*Supplement.* The supplement related to this article is available online

*Competing interests.* Aerosol Dynamics Inc. holds a patent on the particle focusing technology.

*Author contributions.* SS, HT, SVH, AEF and LRW designed the experiments. MA, KT, LRW,
PC, TH, AEF, SRS, and GSL conducted measurements in laboratory and field. Data analysis and
interpretation of the measurement data was done by SS, LRW, AEF and SVH. Working
environment and financial support was provided by HT at FMI, JTJ and DRW at Aerodyne and
SVH at Aerosol Dynamics. SS, LRW and SVH prepared the manuscript with contributions from
all co-authors.

*Acknowledgements.* Funding is gratefully acknowledged from the US Department of Energy,
Small Business Research Program (grant # DESC0004698), the Cityzer (Business Finland project





Dnro:3021/31/2015), TAQIITA (Business Finland project Dnro:2634/31/2015) and the Launching
Regional Innovations and Experimentations Funds (AIKO), governed by the Helsinki Regional
Council (project HAQT, AIKO014).

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



**Table 1**. Approximate temperature and flow settings for the ADIc experiments presented in this study. ADI = Aerosol Dynamics Inc.,
ARI = Aerodyne Research, Inc., FMI = Finnish Meteorological Institute. Tcon, Tini, Tmod and Tnoz are the operating temperatures for
the conditioner, initiator, moderator and focusing nozzle, respectively. AN, AS, DOS are abbreviations for ammonium nitrate,
ammonium sulfate and dioctyl sebacate, respectively.

| Test site | ADI | ADI | ADI | ARI | ARI | FMI | FMI | FMI |
|---|---|---|---|---|---|---|---|---|
| Prototype No. | 1 | 2 | 2 | 1 | 1 | 2 | 2 | 2 |
| Test type | Lab | Lab | Lab | Lab | Field | Lab | Field | Field |
| Measured parameters/ species | Particle number and size | Particle number | Particle number and size | AN, AS | Chemical composition and size | AN, AS, DOS and particle size | Chemical composition | Chemical composition, size |
| Tcond ( °C) | 5 | 5 | 6 | 5 | 5 | 6 | 10 | 10 |
| Tinit ( °C) | 26 | 26 | 31 | 26 | 26 | 31 | 31 | 31 |
| Tmod ( °C) | 10 | 10 | 8 | 10 | 10 | 8 | 13 | 13 |
| Tnoz( °C) | 30 | 30 | 35 | 30 | 30 | 35 | 35 | 35 |
| Tout ( °C) | 35 | 35 | 35 | n/a | n/a | 35 | 35 | 35 |
| Sample Flow (L min$^{-1}$) | 1.0 | 1.0 | 1.5 | 0.9 | 0.9 | 1.7 | 1.0 | 1.7 |
| Output Flow (L min$^{-1}$) | 0.12 | 0.11 | 0.11 | 0.08 | 0.08 | 0.08 | 0.08 | 0.08 |
| Theoretical  CF | 8.3 | 9.1 | 13.6 | 11.3[a] / 12.6[b] | 11.3 | 21.3 | 12.5 | 21.3 |

[a] AN, [b] AS



**Table 2**. Measured and theoretical concentration factors (CFs) for ammonium nitrate (AN) and ammonium sulfate (AS) obtained in the laboratory tests.

| Material | Measured species | Measured CF | Theoretical CF | Measured/ Theoretical CF |
|---|---|---|---|---|
| AS | Particle number | 7.4 | 8.3 | 0.89 |
| | Particle number | 11.9 | 13.6 | 0.88 |
| | Ammonium | 11.2 | 12.6 | 0.89 |
| | Sulfate | 11.3 | 12.6 | 0.89 |
| AN | Ammonium | 10.6 | 11.3 | 0.94 |
| | Nitrate | 10.6 | 11.3 | 0.94 |

**Table 3.** Measured and theoretical concentration factors, and average mass loadings in ambient measurements at ARI. The measured CF was calculated from the ratio of Q-AMS+ADIc to HR-AMS mass loadings. In the bypass line the sample was not concentrated. The theoretical CF was
565 calculated from the ADIc discard flow rate and the Q-AMS inlet flow rate (see text for details).

| | | Through ADIc | Bypass |
|---|---|---|---|
| **Measured CF** | **Organics** | $6.1 \pm 0.8$ | $0.7 \pm 0.06$ |
| | **Sulfate** | $9.7 \pm 1.5$ | $1.0 \pm 0.1$ |
| | **Nitrate** | $9.1 \pm 1.1$ | $1.0 \pm 0.1$ |
| | **Ammonium** | $12.7 \pm 1.9$ | $1.3 \pm 0.4$ |
| **Theoretical CF** | | $10.5 \pm 0.3$ | 1.0 |





Figures

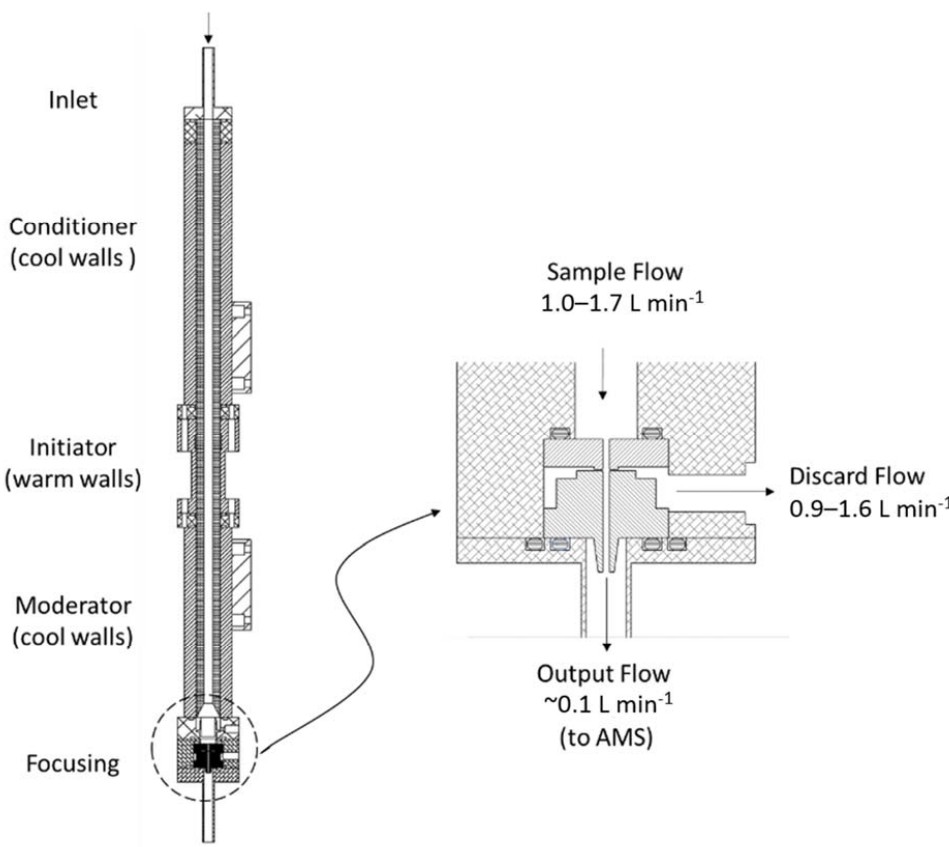

**Figure 1.** Schematic of the Aerosol Dynamics Inc. concentrator (ADIc) with enlargement of the
focusing nozzle.



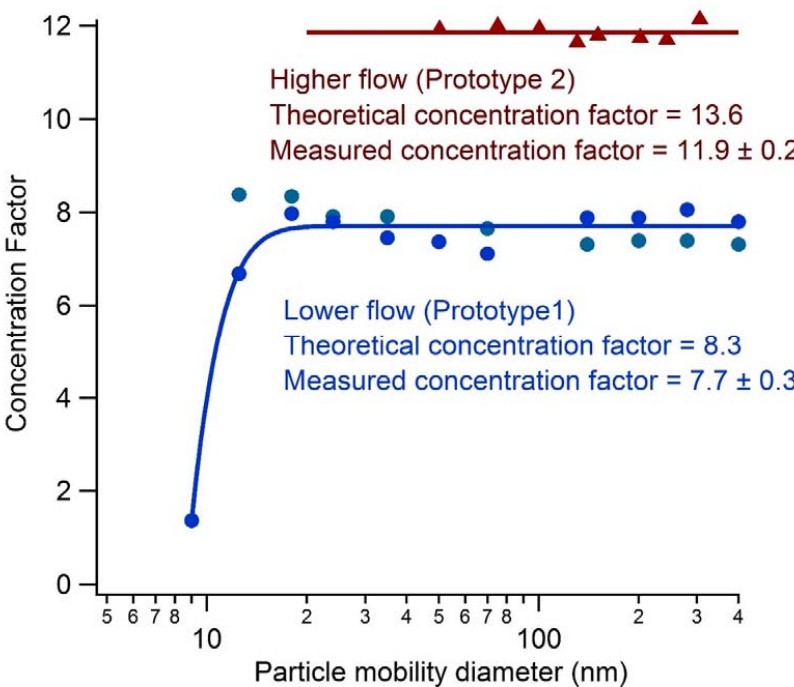

**Figure 2.** Size dependent concentration factor for the ADIc for higher (triangles) and lower (circles) flow regimes as a function of particle size. The red line indicates the average of the higher flow data. The blue line is a guide for the eye. Data are from two different prototype instruments, as indicated.





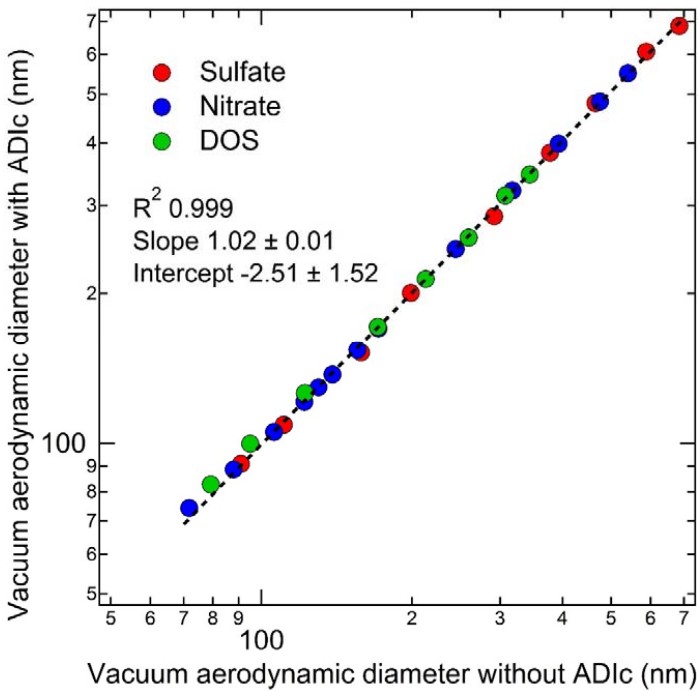

**Figure 3.** Particle size measured with an SP-AMS for 70–700 nm particles (vacuum aerodynamic diameter) of sulfate, nitrate and organics (from DOS) with and without concentration by the ADIc. Corresponding mobility diameters were 30–340 nm.


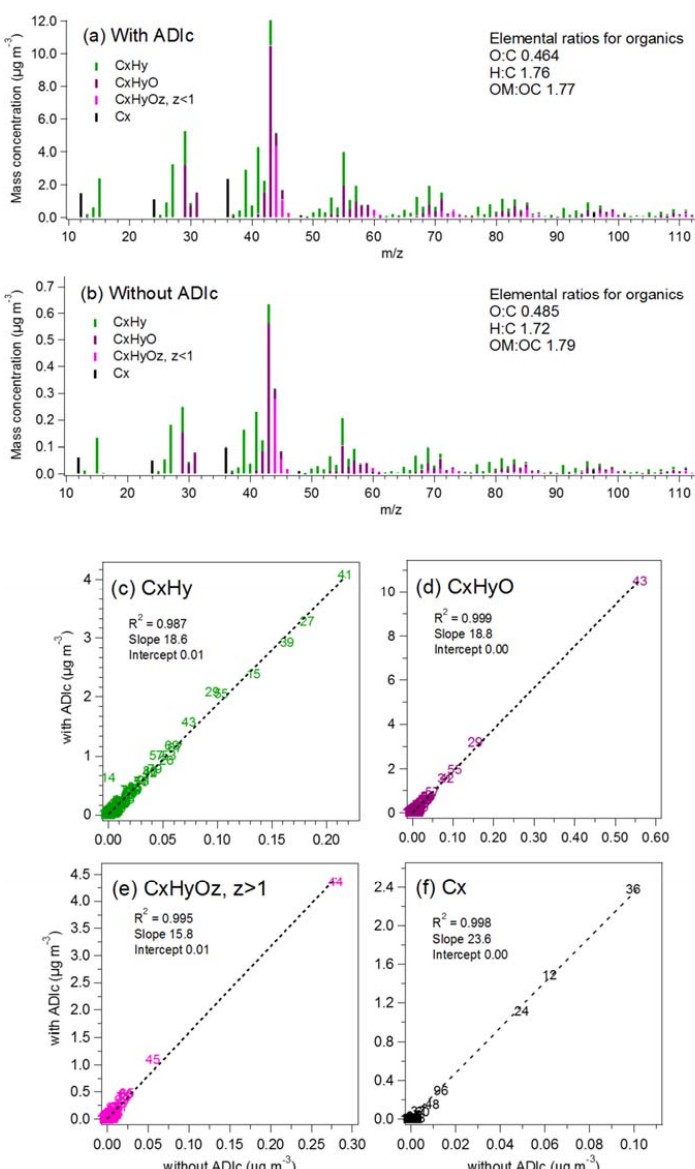

**Figure 4.** Mass spectra for ambient organics and rBC measured with and without ADIc (a–b) and the correlation of AMS fragment families (c–f) at SMEAR III, Helsinki. Theoretical concentration factor was 21.3.





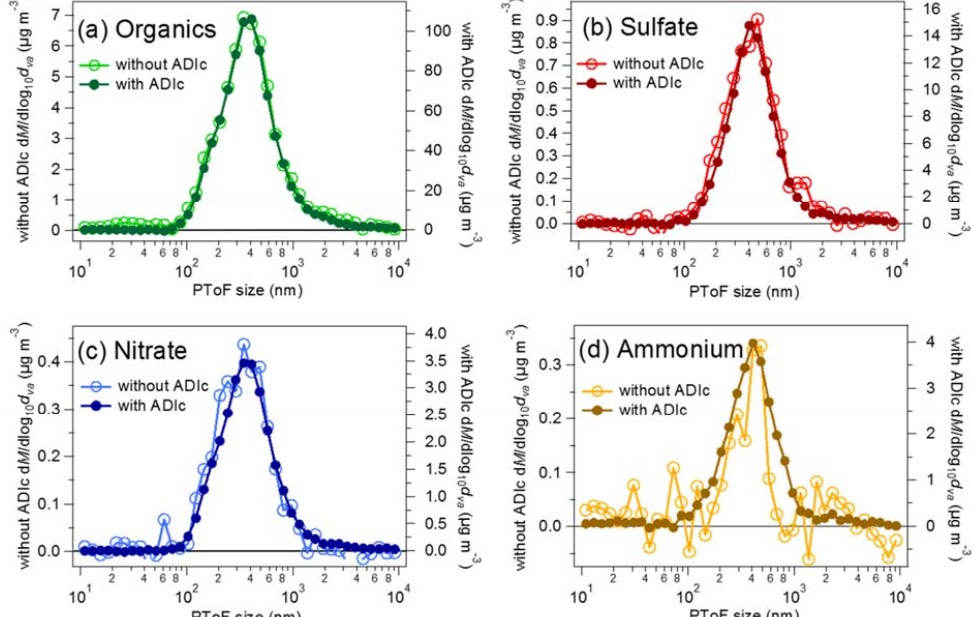

**Figure 5.** Mass size distributions measured without (left axis) and with (right axis) the ADIc for organics (a), sulfate (b), nitrate (c) and ammonium (d) in UMR mode at SMEAR III. Sampling time for each size distribution was 70 minutes with the ADIc and 70 minutes without the ADIc. The theoretical concentration factor was 21.3.





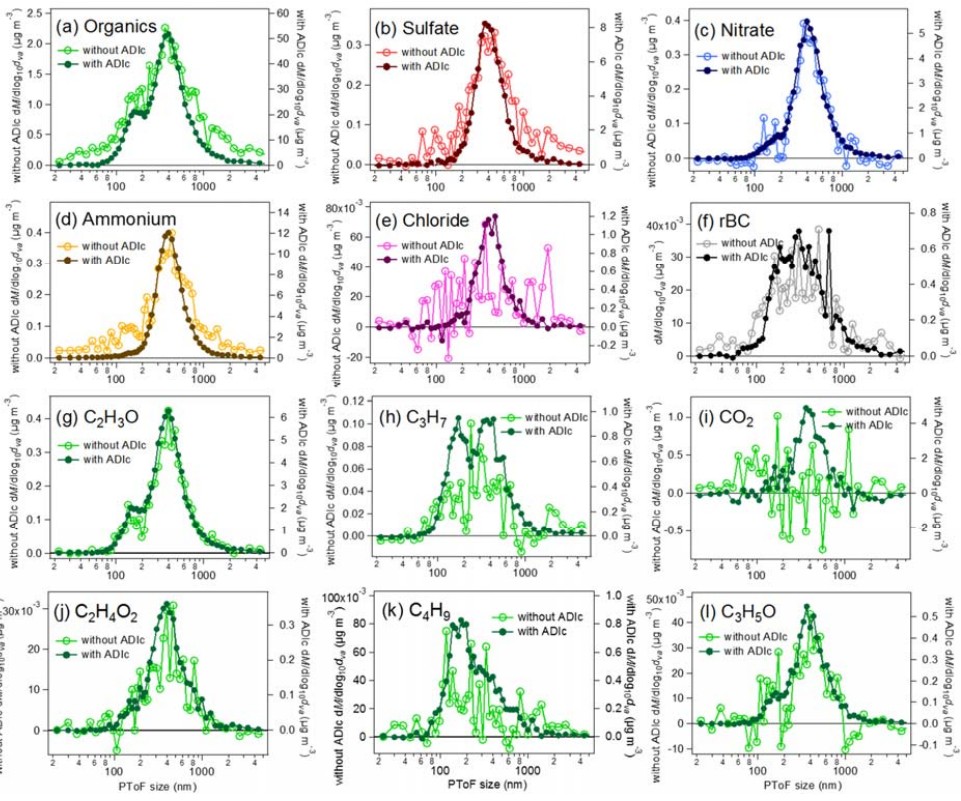

**Figure 6.** Mass size distributions measured without (left axis) and with the ADIc (right axis) for organics (a), sulfate (b), nitrate (c), ammonium (d), chloride (e), rBC (f), $C_2H_3O$ (g), $C_3H_7$ (h), $CO_2$ (i), $C_2H_4O_2$ (j), $C_4H_9$ (k) and $C_3H_5O$ (l) in HR mode at SMEAR III. Sampling time for each size distribution was 70 minutes without and 70 minutes with the ADIc. Theoretical concentration
factor was 21.3.

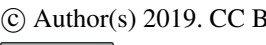



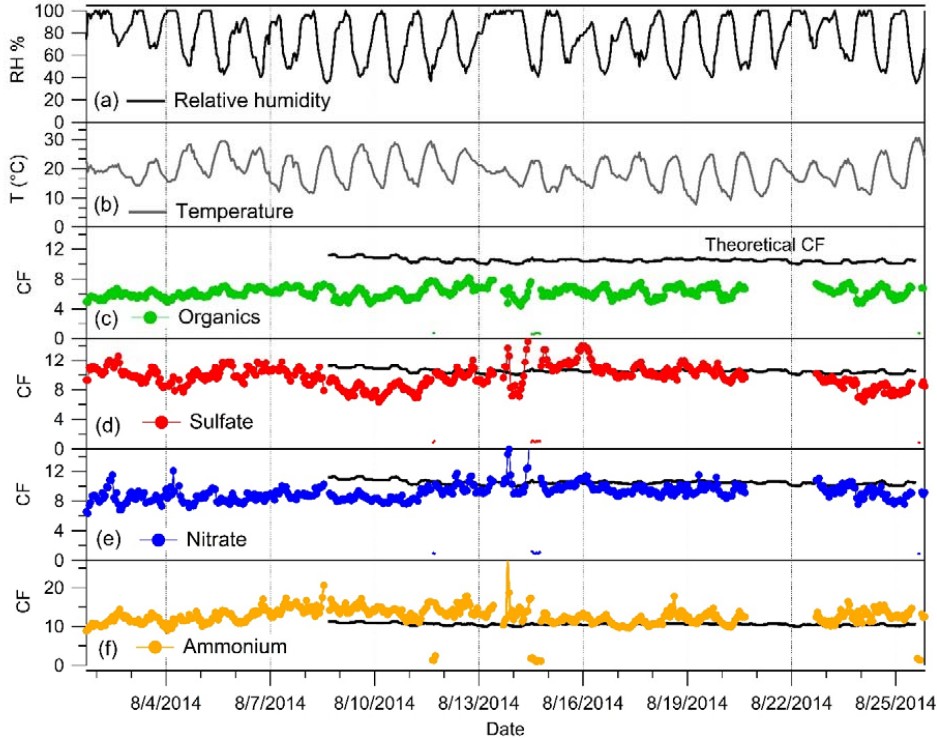

**Figure 7.** Ambient measurements at ARI showing ambient relative humidity (a), ambient
temperature (b) and measured CFs for organics (c), sulfate (d), nitrate (e), and ammonium (f). The
theoretical CF is shown with the black line in (c) – (f).



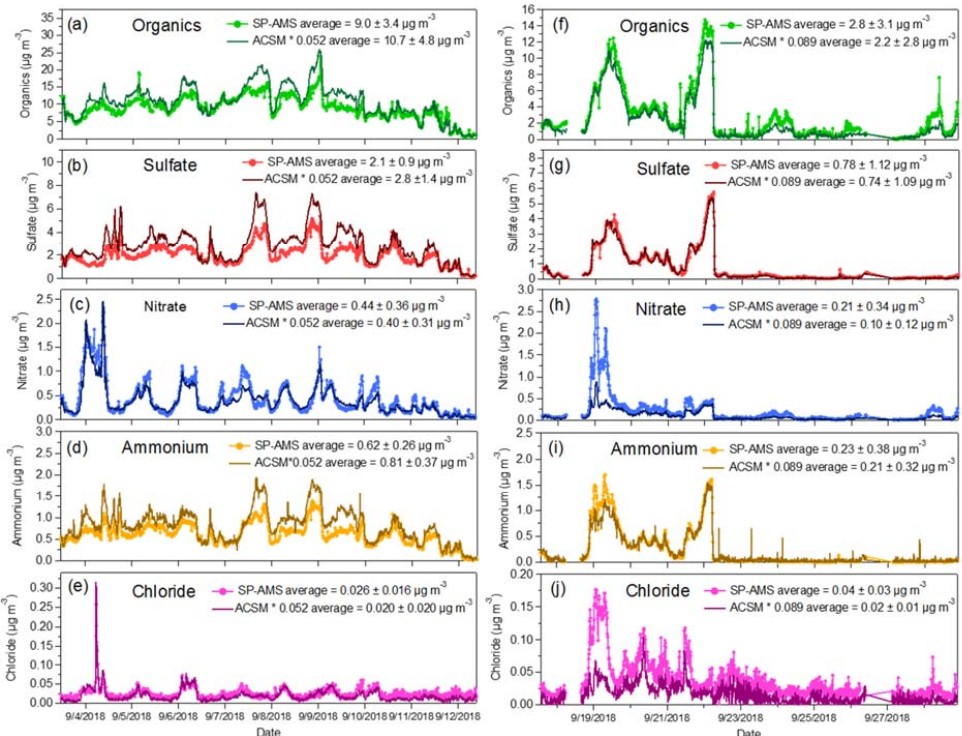

**Figure 8.** Ambient measurements at SMEAR III showing the mass loadings for organics (a, f), sulfate (b, g), nitrate (c, h), ammonium (d, i), and chloride (e, j) measured with the SP-AMS and the ACSM+ADIc in high flow (a–e) and low flow (f–j) regimes. ACSM+ADIc data was corrected for CF as described in the text.





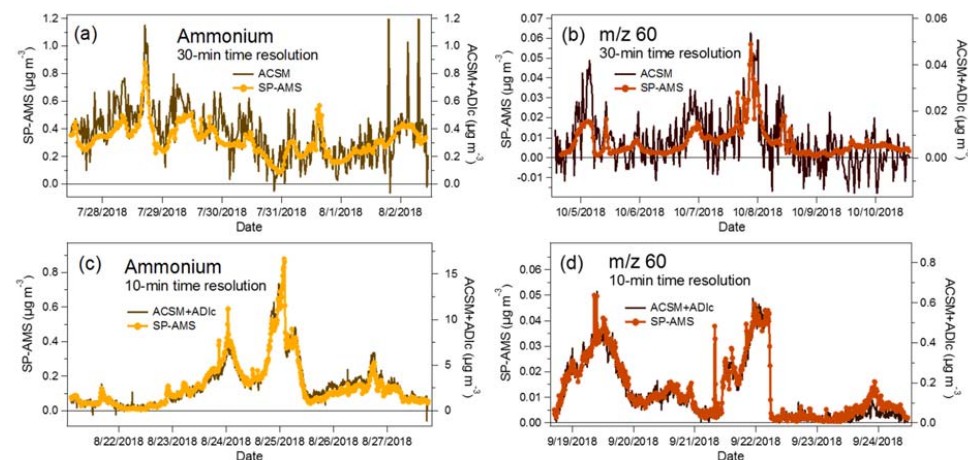

**Figure 9.** Time series of ammonium and m/z 60 with 30-min time resolution with ACSM and SP-AMS (a-b) and 10-min time resolution with SP-AMS and ACSM+ADIc (c)-(d) at SMEAR III
