# Peer review of "Laboratory and field evaluation of the Aerosol Dynamics Inc. concentrator (ADIc)"

_Atmospheric Measurement Techniques, 2019_

## Referee Comment (RC1) · Anonymous Referee #1 · 16 May 2019

This paper introduces ADIc – a particle concentrator designed to increase particle concentrations for sampling with Aerodyne AMS or ATOFMS. Through both lab experiments and field deployments, the authors showed that ADIc can achieve a theoretical concentration factor of 8 − 21 for particles with long-term stability, indicating that ADIc can be a very useful device for enhancing chemical characterization of particles by real-time aerosol mass spectrometry, especially in clean environments where aerosol signals are usually low and frequently close to the instrument detection limits. This work is of high quality and the manuscript is logically organized and generally well written. The topic is a good fit for AMT and I recommend the manuscript be accepted for publication following attention to several issues.

[Figure]

More details on the physical aspects of the ADIc may need to be reported. For example, it would be helpful to know the dimensions of the ADIc growth tube and the residence time of particles for a certain flow rate. Also, for the sake of clarity, consider to add on Figure 1 references to the parameters reported in Table 1. In mentioning the importance of minimizing the time the particle being a droplet inside the growth tube (line 113), it would be useful to quote the approximate time scale. In addition, the issue whether the ADIc modifies the shape or phase of particles should be addressed, at least briefly. Such changes could significantly affect aerosol quantification by the AMS.

Detailed comments:

Line 18, change "ultrafine" to "fine" since ADIc can clearly concentrate particles beyond the ultrafine mode.

Figure 2 shows the size dependent concentration factors for particles only up to 400 nm in mobility diameter. What are the concentration factors for larger particles? Also, the blue circles appear to show in two different shades. Are these from two separate sets of experiments? If so, explain the differences.

Line 201, the sentence "... measured size distributions were normalized to the mass spectra" is vague. Consider to revise.

Line 237, how often was SP-AMS switching between laser-on and laser-off?

For the evaluation of ADIc's influence on aerosol composition and size, Figure 4 is presented to compare the average high resolution mass spectra for organics and rBC from an SP-AMS downstream and bypass the ADIc. The measured-CF for Cx was significantly higher than for the other ions. Could it be due to change in particle shape, thus particle collection efficiency in the laser beam? It would be also interesting to see an evaluation of the ADIc's influence on bulk PM composition, including both inorganics and organics.

Line 311 – 313, this sentence is a bit confusing. Consider to revise.

[Figure]

Line 355 – 358, does it mean that the Q-AMS and the SP-AMS report different am­monium concentration for the same air mass? Won't this discrepancy correctable through proper relative ionization efficiency calibration and fragmentation table adjust­ment (e.g., for better ammonium quantification)?

Fig 8, the ammonium measurement after ADIc shows more spikes. Is this an artifact induced by the ADIc?

[Figure]

---

## Referee Comment (RC2) · Anonymous Referee #2 · 18 May 2019

This paper presents the design, laboratory and field tests of an ultrafine particle concentrator (ADIc) for enhancement of ambient aerosol characterization by aerosol mass spectrometry. The ADIc can concentrate aerosol samples with a theoretical concentration factor ranging from 8 to 21. Laboratory tests show that the ADIc is effective for particles with diameter greater than 10 nm, and the actual concentration factor is close to the theoretical limit. The ADIc does not change the size distribution of ambient aerosol particles, and the impact on aerosol composition is minor. Field tests shows the ADIc is very robust and can run unattended over an extended period. Overall the paper is nicely written, and the topics fits AMT well. I recommend publication of the paper after the authors address the following comments.

[Figure]

(1) Lines 34-35, The sentence "...did not change the size distribution or the chemistry of the ambient aerosol particles." is too strong. The results do suggest there are some minor changes to the particle chemical composition (due to the composition dependence of concentration factor).

(2) Please add diagrams illustrating the setup of the laboratory and field tests (at least in the supplementary information).

(3) Please clarify what a "multiplex chopper" is.

(4) Line 270: how was CF measured? Fig. S2a-b shows the CF was 6.8 instead of 5.7.

(5) Figure S2c-d, the values of regression slope listed in Figures S2c and S2d are different (9.7 and 10.4).

(6) Line 302, how frequently was the sampling alternated between ADIc and the bypass line?

(7) Lines 368-369: The low particle transmission efficiency through the lens is unlikely the only cause for the low CF. Figure S3c shows that in the lower size range (e.g., 400-600 nm), the CF was about 5, substantially below the theoretical value. How do the measurements of Q-AMS with ADIc bypassed compare with HR-AMS data for different species?
* * *

---

## Author Comment (AC3) · 22 Jun 2019

Addition to Authors′ response to Referee #2 comments Comment (3) Please clarify what a "multiplex chopper" is. Response: In the AMS, the transmission of the beam to the particle detector is modulated with a mechanical chopper that is operated at 100–150 Hz. Time-resolved detection of the particles, coupled with the known flight distance, gives the particle velocity from which the particle aerodynamic diameter is obtained. Typically, the chopper wheel has one or two radial slits giving a sampling duty cycle of 1–4%. The multiplex chopper has multiple slits in a specific sequence, such that particles of many sizes are arriving at the detector at any given time. This

multiplexed signal is then deconvolved with a Hadamard transform to retrieve the particle size distribution. The advantage is that the particle throughput is close to 50% leading to better signal to noise. We call this the efficient Particle time of Flight (ePToF) chopper. Unfortunately, we don't have a reference for this yet. Author's changes in manuscript: We added to the text: "were equipped with a multiple slit chopper (efficient Particle Time of Flight, ePToF, chopper) with 50% particle throughput"
* * *

---

## Author Response (AR1)

Dear editor,

We thank the two Referees for their valuable comments on our manuscript "Laboratory and field evaluation of the Aerosol Dynamics Inc. concentrator (ADIc) for aerosol mass spectrometry". We think that the revision we made to the manuscript, based on the comment of the Reviewers, has improved the quality of the manuscript significantly. Most of the changes suggested by the Referees were implemented by adding requested technical details to the manuscript but we also added two figures to Supplemental material. In addition, some minor changes has been made throughout the manuscript to improve the grammar. All the changes to the manuscript have been made by "Track changes" mode and the changes in Supplemental information have been highlighted in yellow. Additionally, point-by-point responses to the comments of the Referees are given in separate author's responses.

On behalf of all authors,

Sanna Saarikoski

Authors´ response to Referee #1 comments

We would like to thank Referee #1 for the valuable comments that aided us to improve the manuscript. In this post, we will provide our response to the Referee's comments. In our replies, we provide original comment from the Referee and our response followed by the changes made to the manuscript.

General comment:

(1) More details on the physical aspects of the ADIc may need to be reported. For example, it would be helpful to know the dimensions of the ADIc growth tube and the residence time of particles for a certain flow rate. Also, for the sake of clarity, consider to add on Figure 1 references to the parameters reported in Table 1. In mentioning the importance of minimizing the time the particle being a droplet inside the growth tube (line 113), it would be useful to quote the approximate time scale. In addition, the issue whether the ADIc modifies the shape or phase of particles should be addressed, at least briefly. Such changes could significantly affect aerosol quantification by the AMS.

**Response and author's changes in manuscript:**

Dimensions: The dimensions of the growth tube are already stated in the manuscript (see the text starting on line 126 (page 5) "The conditioner, initiator and moderator are 140 mm, 51 mm and 102 mm long, respectively, separated by 7.5 mm thick insulator sections. In both prototypes the growth tube was lined with a 9 mm-ID, ~1.5 mm-thick wick formed from rolled membrane filter."

Residence Time: We added to the manuscript: "For particles along the centerline of the flow, the calculated residence time from the point of activation to the inlet of the focusing nozzle is 200-300 ms, depending on the point of activation. Along the flow trajectory that encompasses 50% of the flow, the residence time is as long as 400 ms."

Operating temperatures for conditioner, initiator, moderator and focusing nozzle have been added to Fig. 1.

Particle Shape: Discussion of particle shape has been added to manuscript (see details in response to comment 6).

Detailed comments:

(2) Line 18, change "ultrafine" to "fine" since ADIc can clearly concentrate particles beyond the ultrafine mode.

**Response and author's changes in manuscript:** "ultrafine" has been changed to "fine"

(3) Figure 2 shows the size dependent concentration factors for particles only up to 400 nm in mobility diameter. What are the concentration factors for larger particles? Also, the blue circles appear to show in two different shades. Are these from two separate sets of experiments? If so, explain the differences.

**Response and author's changes in manuscript:** Unfortunately we were not able to investigate the concentration factors for the particles larger than 400 nm (in mobility diameter) in the laboratory due to the instrumentation available. However, based on the ambient size distribution data measured by the SP-AMS at SMEAR III (Fig. 5), the CFs were rather stable until 1 µm in vacuum diameter (dva) corresponding to ~600 nm in mobility diameter (see figure below).

[Figure]

**Fig. R1.** Size dependent concentration factor for nitrate, sulfate and organics during the field measurements at SMEAR III in Helsinki.

In contrast, during the field measurements at ARI, the size distributions for organics and m/z 57 from the Q-AMS+ADIc were missing mass above dva ~ 700 nm that was measured by the HR-AMS without the ADIc. This difference can be at least partly explained by a difference in the cutoff of the aerodynamic lenses in the two AMS instruments. The difference in the size distributions is discussed also in comment (7) for Referee #2.

All blue circles are from the same experiments so they should appear in same color.

No changes were made to manuscript based on this comment.

(4) Line 201, the sentence ". . . measured size distributions were normalized to the mass spectra" is vague. Consider to revise.

> **Response and author's changes in manuscript:** The sentence was modified to: "The measured size distributions were normalized to the mass concentrations measured in the mass spectrum mode."

(5) Line 237, how often was SP-AMS switching between laser-on and laser-off?

> **Response and author's changes in manuscript:** Laser was switching between on and off every 1.5 minutes. Switching period has been added to manuscript.

(6) For the evaluation of ADIc's influence on aerosol composition and size, Figure 4 is presented to compare the average high resolution mass spectra for organics and rBC from an SP-AMS downstream and bypass the ADIc. The measured-CF for Cx was significantly higher than for the other ions. Could it be due to change in particle shape, thus particle collection efficiency in the laser beam? It would be also interesting to see an evaluation of the ADIc's influence on bulk PM composition, including both inorganics and organics.

> **Response and author's changes in manuscript:** The influence of ADIc on particle shape has been added to the text: "One possible explanation is that the ADIc altered the shape of the rBC-containing particles. The effect of the condensation/evaporation process on particle shape was not explored in this study; however, others have found changes in the shape of aggregates. In a study using a condensation system similar to that employed here, Ma et al (2013) reported collapse of the aggregate structure of laboratory-generated soot in the evaporation process. Regarding the SP-AMS, the morphology of the particles had been demonstrated to affect the collection efficiency since it affects the overlap of the particle beam and the laser beam (Willis et al., 2014)."

The bulk PM composition measured by the SP-AMS with and without the ADIc has been added to Supplemental material (Fig. S4).

References

Ma, X., Zangmeister, C. D., Gigault, J., Mulholland, G. W., & Zachariah, M. R. (2013). Soot aggregate restructuring during water processing. *Journal of Aerosol Science*, *66*, 209-219.

Willis, M.D., Lee, A.K.Y., Onasch, T.B., Fortner, E.C., Williams, L.R., Lambe, A.T., Worsnop, D.R., Abbatt, J.P.D., 2014. Collection efficiency of the soot-particle aerosol mass spectrometer (SP-AMS) for internally mixed particulate black carbon. Atmos. Meas. Tech. 7, 4507–4516.

(7) Line 311 – 313, this sentence is a bit confusing. Consider to revise.

**Response and author's changes in manuscript:** That sentence has been modified as well as few other sentences related to it.

(8) Line 355 – 358, does it mean that the Q-AMS and the SP-AMS report different ammonium concentration for the same air mass? Won't this discrepancy correctable through proper relative ionization efficiency calibration and fragmentation table adjustment (e.g., for better ammonium quantification)?

**Response and author's changes in manuscript:** The referee is correct that we did not have good agreement in bypass for ammonium between the Q-AMS and the HR-AMS, even with several RIE calibrations and adjustments to the fragmentation tables. Part of the problem is that ammonium concentration was low (< 0.4 ug m$^{-3}$), and it was often close to the detection limit for the Q-AMS during the bypass periods. We think that this is an indication of how hard it is to get two instruments to agree for all species.

Lines 355-358 were revised:

"Another possibility is that the RIE for ammonium was incorrect for one or both of the instruments, even though it was measured before and after the ambient sampling period with pure AN particles. The CF during bypass periods was 1.3 ± 0.4 (Table 3) indicating that the two instruments did not agree well for ammonium even when the Q-AMS was bypassing the ADIc. However, the ammonium mass loading was low (<0.4 ug m$^{-3}$) and often close to the detection limit for the Q-AMS during the bypass periods, leading to a large uncertainty in the bypass CF."

(9) Fig 8, the ammonium measurement after ADIc shows more spikes. Is this an artifact induced by the ADIc?

**Response and author's changes in manuscript:** Ammonium spikes in the time series of the ACSM are not induced by the ADIc since similar spikes are seen in Fig. 9a when the ACSM was used in bypass without the ADIc. We think that these spikes are likely related to the detection of small air bubbles in the ACSM that affect the measured ammonium concentration. The spikes may be either negative or positive if the air bubble is released during the filter or aerosol measurement phase.

We added to figure caption 8: "Spikes in the time series of ammonium in the ACSM are likely related to the detection of small air bubbles in the ACSM that affect the measured ammonium concentration."

Authors´ response to Referee #2 comments

We would like to thank Referee #2 for the constructive comments that helped us to improve the manuscript. In this post, we will provide our response to the Referee's comments. In our replies, we provide original comment from the Referee and our response followed by the changes made to the manuscript.

(1) Lines 34-35, The sentence ". . .did not change the size distribution or the chemistry of the ambient aerosol particles." is too strong. The results do suggest there are some minor changes to the particle chemical composition (due to the composition dependence of concentration factor).

>    **Response and author's changes in manuscript:** sentence is now …did not significantly change the size distribution…

(2) Please add diagrams illustrating the setup of the laboratory and field tests (at least in the supplementary information).

>    **Response and author's changes in manuscript:** Set-ups for the laboratory and fields test have been added to Supplemental information (Fig. S2).

(3) Please clarify what a "multiplex chopper" is.

>    **Response and author's changes in manuscript:** multiplex chopper is an efficient Particle time of Flight (ePToF) chopper that is based on a multiplexed particle beam chopper system with 50% particle throughput  providing significantly improved signal-to-noise for the particle size measurement (compared to standard 1–2% throughput). We added to the text: …"(efficient Particle Time of Flight, ePToF, chopper) with 50% particle throughput."

(4) Line 270: how was CF measured? Fig. S2a-b shows the CF was 6.8 instead of 5.7.

>    **Response and author's changes in manuscript:** The CF of 5.7 was an average of the CFs calculated separately to each data point (n=652) while the CF based on the regression slope was 6.8.  We think that the average of the CFs is a better representation of the data since the regression slope can be biased by the large values. In addition, it gives a more realistic uncertainty.  We have changed the text to read:
>
>    **"For the lower flow regime data (Fig. S3a–b), the average CF, calculated as the ratio of the number concentration in the output flow to that in the sample flow, was 5.7 ± 0.4 with a theoretical CF of 7.5. Linear regression of that data yielded a correlation coefficient ($R^2$) of 0.984. In the higher flow regime (Fig. S3c–d), the measured CF was 9.0 ± 0.7, with a theoretical CF of 13.6."

(5) Figure S2c-d, the values of regression slope listed in Figures S2c and S2d are different (9.7 and 10.4).

>    **Response and author's changes in manuscript:** The correct regression slope in Fig. S2c (now Fig. S3c) is 10.4. The figure has been changed.

(6) Line 302, how frequently was the sampling alternated between ADIc and the bypass line?

>    **Response and author's changes in manuscript:** The SP-AM was switching between the bypass line and the ADIc every 30 minutes. Switching period has been added to the text.

(7) Lines 368-369: The low particle transmission efficiency through the lens is unlikely the only cause for the low CF. Figure S3c shows that in the lower size range (e.g., 400- 600 nm), the CF was about 5, substantially below the theoretical value. How do the measurements of Q-AMS with ADIc bypassed compare with HR-AMS data for different species?

**Response and author's changes in manuscript:** The referee is right that the CF for organics and m/z 57 (~6 and ~4, respectively) was much lower than the theoretical CF (10.5) at the size range of 400–600 nm. Unfortunately the bypass period was rather short and the Q-AMS size distribution data was too noisy to be compared with the size distributions from the HR-AMS during the bypass.

However, in Table 3 we present the ratio of Q-AMS to HR-AMS mass loadings (without size distribution information) for the chemical species during the bypass period. The mass concentrations from the Q-AMS and HR-AMS in bypass agreed for sulfate and nitrate while ammonium had larger concentrations from the Q-AMS in bypass (for ammonium see comment (8) for Referee #1). In terms of organics, the mass loadings measured by the Q-AMS were smaller than those from the HR-AMS in bypass (ratio=0.7). This suggests that the low CF for organics can be partly due to the fact that the two instruments did not agree well for organics even when the Q-AMS was bypassing the ADIc. To investigate this difference, the mass spectra of organics from the Q-AMS with the ADIc and in bypass was compared to the mass spectra from the HR-AMS (in bypass) in the unit mass resolution mode (see Figure below). It is clear that m/z 44 agrees pretty well for the two instruments but the HR-AMS has more signal at most m/z's, especially at higher m/z's. It's possible that there was more fragmentation in the Q-AMS, but it's also possible that there was always road paving aerosol in the air and the lens cutoff affected the mass spectra even during bypass.

**(a) Mass spectra for organics for HR-AMS and Q-AMS with the ADIc**

[Figure]

**(b) Mass spectra for organics for HR-AMS and Q-AMS without the ADIc**

[Figure]

**Fig. R1.** Mass spectra for organics measured with the Q-AMS with the ADIc and HR-AMS in bypass (without the ADIc) (a), and the Q-AMS and HR-AMS without the ADIc (b).

Nevertheless, it can't be ruled out totally, that the concentration process was less effective for hydrocarbon-like organics than for e.g. sulfate during the field test at ARI. However, during the measurements in Helsinki, just the opposite was found. At SMEAR III hydrocarbon-like organics had higher CF than highly oxygenated organics (Fig. 4).

We added to manuscript: "Besides the lens cut-off, it is possible that the CF was smaller for hydrocarbon-like organics than for oxygenated organics during the measurements at ARI. However, that is just the opposite of what was found at SMEAR III in Helsinki where hydrocarbon-like fragment ions had higher CF than highly oxygenated fragment ions (Fig. 4)."

We added two sentences about the agreement between the two instruments during bypass:

"Average values of CF are presented in Table 3, along with the ratio of the mass loadings during bypass periods." in the first paragraph of Section 3.2.3 and "The agreement between the two instruments during bypass periods was excellent for nitrate and sulfate (Table 3)." in the second paragraph.

Figure S5a has been changed because it contained incorrect data.

Also, "the average mass loadings" have been removed from the caption for Table 3 because the mass loadings were not presented in Table 3.

[revised manuscript text omitted]

Figures

[Figure]

Inlet

Conditioner
(5–10 °C)

Initiator
(26–31 °C)

Moderator
(8–13 °C)

Focusing
(30–35 °C)

Sample flow
1.0–1.7 L min$^{-1}$

Discard flow
0.9–1.6 L min$^{-1}$

Output flow
~0.1 L min$^{-1}$
to AMS

[Figure]

**Figure 1.** Schematic of the Aerosol Dynamics Inc. concentrator (ADIc) with enlargement of the focusing nozzle.

[Figure]

**Figure 2.** Size dependent concentration factor for the ADIc for higher (triangles) and lower (circles) flow regimes as a function of particle size. The red line indicates the average of the higher flow data. The blue line is a guide for the eye. Data are from two different prototype instruments, as indicated.

[Figure]

**Figure 3.** Particle size measured with an SP-AMS for 70–700 nm particles (vacuum aerodynamic diameter) of sulfate, nitrate and organics (from DOS) with and without concentration by the ADIc. Corresponding mobility diameters were 30–340 nm.

[Figure]

**Figure 4.** Mass spectra for ambient organics and rBC measured with and without ADIc (a–b) and the correlation of AMS fragment families (c–f) at SMEAR III, Helsinki. Theoretical concentration factor was 21.3.

[Figure]

**Figure 5.** Mass size distributions measured without (left axis) and with (right axis) the ADIc for organics (a), sulfate (b), nitrate (c) and ammonium (d) in UMR mode at SMEAR III. Sampling time for each size distribution was 70 minutes with the ADIc and 70 minutes without the ADIc. The theoretical concentration factor was 21.3.

[Figure]

**Figure 6.** Mass size distributions measured without (left axis) and with the ADIc (right axis) for organics (a), sulfate (b), nitrate (c), ammonium (d), chloride (e), rBC (f), $C_2H_3O$ (g), $C_3H_7$ (h), $CO_2$ (i), $C_2H_4O_2$ (j), $C_4H_9$ (k) and $C_3H_5O$ (l) in HR mode at SMEAR III. Sampling time for each size distribution was 70 minutes without and 70 minutes with the ADIc. Theoretical concentration factor was 21.3.

[Figure]

**Figure 7.** Ambient measurements at ARI showing ambient relative humidity (a), ambient temperature (b) and measured CFs for organics (c), sulfate (d), nitrate (e), and ammonium (f). The theoretical CF is shown with the black line in (c) – (f).

[Figure]

**Figure 8.** Ambient measurements at SMEAR III showing the mass loadings for organics (a, f), sulfate (b, g), nitrate (c, h), ammonium (d, i), and chloride (e, j) measured with the SP-AMS and the ACSM+ADIc in high flow (a–e) and low flow (f–j) regimes. ACSM+ADIc data was corrected for CF as described in the text. Spikes in the time series of ammonium in the ACSM are likely related to the detection of small air bubbles in the ACSM that affect the measured ammonium concentration.

[Figure]

**Figure 9.** Time series of ammonium and m/z 60 with 30-min time resolution with ACSM and SP-AMS (a-b) and 10-min time resolution with SP-AMS and ACSM+ADIc (c)-(d) at SMEAR III

**Laboratory and field evaluation of the Aerosol Dynamics Inc. concentrator (ADIc) for aerosol mass spectrometry**

Sanna Saarikoski[1], Leah R. Williams[2], Steven R. Spielman[3], Gregory S. Lewis[3], Arantzazu Eiguren-Fernandez[3], Minna Aurela[1], Susanne V. Hering[3], Kimmo Teinilä[1], Philip Croteau[2], John T. Jayne[2], Thorsten Hohaus[2,+], Douglas R. Worsnop[2], Hilkka Timonen[1]

[1] Atmospheric Composition Research, Finnish Meteorological Institute, Helsinki, Finland

[2] Center for Aerosol and Cloud Chemistry, Aerodyne Research, Inc., Billerica, MA, USA

[3] Aerosol Dynamics Inc., Berkeley, CA, USA

[+] Now at Institute of Energy and Climate Research, IEK-8: Troposphere, Forschungszentrum Juelich GmbH, Juelich, Germany

Supplemental Information

[Figure]

**Figure S1.** Calculated particle trajectories for different particle sizes entering the focusing nozzle of the ADIc. Scale is expanded radially for better visualization.

**ADI LABORATORY TESTS**

[Figure]

**ADI LABORATORY AIR**

[Figure]

**ARI LABORATORY TESTS**

[Figure]

**ARI FIELD TESTS**

[Figure]

**FMI LABORATORY TESTS**

[Figure]

**FMI FIELD TESTS WITH Q-ACSM AND SP-AMS**

[Figure]

**FMI FIELD TESTS WITH SP-AMS**

[Figure]

**Figure S2.** Diagrams for the instrumental set-ups used in the laboratory and field tests at Aerosol Dynamics Inc. (ADI), Aerodyne Research, Inc. (ARI) and Finnish Meteorological Institute (FMI).

[Figure]

**Figure S3.** Particle number concentrations in the ADIc sample and output flows while sampling laboratory air shown as time series (a, c) and as correlation plots (b, d). Prototype 1 was operating at low flow (a–b) and prototype 2 at high flow (c–d).

[Figure]

**Figure S4.** Chemical composition of particles with the ADIc (a) and without the ADIc (b) measured with the SP-AMS at SMEAR III. Sampling time was 70 minutes with the ADIc and 70 minutes without the ADIc. The theoretical concentration factor was 21.3.

[Figure]

**Figure S5.** Size distributions for organics (a), m/z 44 (b) and m/z 57 (c) from the HR-AMS in bypass (without the ADIc) and the Q-AMS behind the ADIc demonstrating different size cutoffs in the aerodynamic lenses >700 nm in the two instruments.

---

## Author Response (AR2)

Dear editor,

Thank you for your comments and the decision about our manuscript "Laboratory and field evaluation of the Aerosol Dynamics Inc. concentrator (ADIc) for aerosol mass spectrometry". Regarding your comment on multiplex chopper:

"I would like you to provide a more comprehensive response to Reviewer 2's question: '(3) Please clarify what a "multiplex chopper" is.' You essentially responded, "it's a chopper that multiplexes." This doesn't really explain what multiplexing means in the context of a beam chopper. Please provide a concise but more descriptive explanation, with a reference if appropriate."

we have revised our response to Reviewer 2's comment (3). New response is:

> **Response**: In the AMS, the transmission of the beam to the particle detector is modulated with a mechanical chopper that is operated at 100–150 Hz. Time-resolved detection of the particles, coupled with the known flight distance, gives the particle velocity from which the particle aerodynamic diameter is obtained. Typically, the chopper wheel has one or two radial slits giving a sampling duty cycle of 1–4%. The multiplex chopper has multiple slits in a specific sequence, such that particles of many sizes are arriving at the detector at any given time. This multiplexed signal is then deconvolved with a Hadamard transform to retrieve the particle size distribution. The advantage is that the particle throughput is close to 50% leading to better signal to noise. We call this the efficient Particle time of Flight (ePToF) chopper. Unfortunately, we don't have a reference for this yet.

> **Changes in manuscript:** We added to the text: "were equipped with a multiple slit chopper (efficient Particle Time of Flight, ePToF, chopper) with 50% particle throughput"

We posted a new authors' comment regarding this revision.

On behalf of all authors,

Sanna Saarikoski